# Biodegradable Antimicrobial Food Packaging: Trends and Perspectives

**DOI:** 10.3390/foods9101438

**Published:** 2020-10-11

**Authors:** Ludmila Motelica, Denisa Ficai, Anton Ficai, Ovidiu Cristian Oprea, Durmuş Alpaslan Kaya, Ecaterina Andronescu

**Affiliations:** 1Faculty of Applied Chemistry and Materials Science, University Politehnica of Bucharest, 060042 Bucharest, Romania; motelica_ludmila@yahoo.com (L.M.); denisaficai@yahoo.ro (D.F.); anton.ficai@upb.ro (A.F.); ecaterina.andronescu@upb.ro (E.A.); 2Section of Chemical Sciences, Academy of Romanian Scientists, 050045 Bucharest, Romania; 3Department of Field Crops, Faculty of Agriculture, Hatay Mustafa Kemal University, 31030 Antakya Hatay, Turkey; dak1976@msn.com

**Keywords:** biodegradable, polymeric nanocomposite, antimicrobial packaging, edible films, chitosan, starch, cellulose, polylactic acid, essential oils, nanoparticles toxicity

## Abstract

This review presents a perspective on the research trends and solutions from recent years in the domain of antimicrobial packaging materials. The antibacterial, antifungal, and antioxidant activities can be induced by the main polymer used for packaging or by addition of various components from natural agents (bacteriocins, essential oils, natural extracts, etc.) to synthetic agents, both organic and inorganic (Ag, ZnO, TiO_2_ nanoparticles, synthetic antibiotics etc.). The general trend for the packaging evolution is from the inert and polluting plastic waste to the antimicrobial active, biodegradable or edible, biopolymer film packaging. Like in many domains this transition is an evolution rather than a revolution, and changes are coming in small steps. Changing the public perception and industry focus on the antimicrobial packaging solutions will enhance the shelf life and provide healthier food, thus diminishing the waste of agricultural resources, but will also reduce the plastic pollution generated by humankind as most new polymers used for packaging are from renewable sources and are biodegradable. Polysaccharides (like chitosan, cellulose and derivatives, starch etc.), lipids and proteins (from vegetal or animal origin), and some other specific biopolymers (like polylactic acid or polyvinyl alcohol) have been used as single component or in blends to obtain antimicrobial packaging materials. Where the package’s antimicrobial and antioxidant activities need a larger spectrum or a boost, certain active substances are embedded, encapsulated, coated, grafted into or onto the polymeric film. This review tries to cover the latest updates on the antimicrobial packaging, edible or not, using as support traditional and new polymers, with emphasis on natural compounds.

## 1. Introduction

The Food and Agriculture Organization of the United Nations, FAO, statistical data indicate that about one-third of the produced food is lost or wasted each year because to shelf life expiring, alteration, or spoilage due to microbial activity. About 40–50% of fruits and vegetables, 35% of fish, 30% of cereals, and 20% of dairy and meat products are lost yearly. The main culprits are the presence of bacteria and fungi, oxygen-driving processes, and presence of some enzymes. In the less developed countries 40% of the total losses occurs right after harvesting, until processing, because of poor infrastructure while in the developed countries 40% of total losses occur at retail and customer level as food waste. At retail level large quantities are wasted because of the artificial emphasis put on the exterior look [1]. Generally speaking, the food loss occurs at production, handling, storage, and processing level, while food waste indicate the losses at distribution, market, and final consumer level [2,3].

The global food packaging market was estimated at over USD 300 billion in 2019 and has a predicted increasing rate of 5.2% yearly. The paper-based segment represents 31.9% of the total value and dominate as a biodegradable material. Unfortunately, the plastic food packaging is expected to exhibit a rapid growth globally due to superior properties and lower prices. This increase comes with environmental and pollution problems [4,5]. In developed countries the market is highly regulated by agencies such as Food and Drug Administration (FDA) or European Commission (EC). Those regulatory bodies are shifting the market toward more sustainable packaging solutions like bio-based polymers and are putting pressure on reduction of traditional plastic packaging. In the same time the economy must be transformed from a linear model to the circular economy [6]. The packaging is one of the most powerful tools in food industry as it acts also as a communication and branding medium beside its traditional role. As a marketing tool the packaging can promote a healthier lifestyle and change by itself the consumer habits [7,8,9,10,11,12].

At present, most food packaging materials are based on petrochemical polymers because of historic factors, low-cost, and good barrier performances. These polymers are nonbiodegradable and worldwide they have already raised a lot of environmental concerns regarding short- and long-term pollution [13,14,15,16,17,18,19]. These two factors are putting an increasing pressure on food industry to develop new types of antimicrobial packaging materials, mainly based on natural, renewable sources, or biopolymers that are environmentally safe. While the simplest way to achieve antimicrobial activity is to modify the currently used packaging materials, the environmental pressure is slowly phasing out the nonbiodegradable polymers [20,21,22,23,24,25].

The development of antimicrobial packages is a promising path for actively controlling the bacterial and fungal proliferation that leads to food spoilage [26,27,28,29]. Antimicrobial packages have a pivotal role in food safety and preservation. This type of package increases the latency phase and reduce the growth of microorganisms, enhancing the food quality and safety, ensuring a longer shelf life [30,31,32,33]. By using antimicrobial materials for food package, the shelf life is extended and the growth of microorganisms is slowed, thus ensuring a better quality and safety for meat, vegetables, fruits, and dairy products [34,35,36,37].

The methods employed in obtaining antimicrobial activity vary from adding sachets with volatile antimicrobial agents inside current packaging to incorporating the active agent directly into biopolymers and from coating or grafting antimicrobial substances on polymeric surface to the use of intrinsic antimicrobial polymers or antimicrobial pads (Figure 1) [38,39,40,41,42].

The antimicrobial packaging strategies can be classified in two types. First type is represented by the packaging materials with direct contact between the antimicrobial surface and the preserved food, in which the active agents can migrate into the food. Such packaging are used for food that is wrapped in foils or under vacuum. A second strategy is having the antimicrobial agent inside the package, but not in direct contact with the food, and here can be mentioned the modified atmosphere packaging (MAP) [26,43,44,45].

Although an increasing number of antimicrobial agents (like ethanol, carbon dioxide, silver ions or nanoparticles, chlorine dioxide, antibiotics, organic acids, essential oils etc.) [30,46,47,48,49] have been tested with the aim of inhibiting the growth of microorganisms on food, the number of antimicrobial packaging commercially available is still limited. A notable exception is in the case of silver-based antimicrobial packaging that are in use in countries like Japan or the United States [44]. The introduction of silver-based antimicrobial package solutions is expected to grow also in E.U. after the inclusion on the provisional list of permitted food additives and in the list of permitted surface biocidal products. The literature reports a wide range of antimicrobial agents like metallic nanoparticles (Ag, Cu, bimetallic etc.) [50,51,52], oxide nanoparticles (ZnO, TiO_2_, CuO, etc.) [53,54,55], clay nanoparticles (cloisite, montmorillonite, bentonite) [56,57,58], natural extracts (essential oils or hydrophilic extracts) [59,60,61], natural antimicrobials (nisin, pediocin, antibiotics, etc.) [62,63,64], biopolymers (chitosan) [65,66,67], enzymes (lysozyme, peroxidase) [68,69,70], synthetic antimicrobial agents (including synthetic antibiotics) [71].

The natural antimicrobial compounds are obtained from vegetal sources like cloves, cinnamon, thyme, ginger, oregano, rosemary, garlic, etc. All these have a good potential for the meat industry [72,73]. In a similar way, other natural antimicrobial agents can be isolated from substances produced by bacterial or fungal activity such as pediocin, nisin, and various bacteriocins. In the meat industry, currently, some technologies based on blocking the microbial activity are in use that can enhance the organoleptic properties and packaging performance [30,74,75,76,77]. Various synthetic antimicrobial agents have been reported for bread and pastry packaging [78,79,80] or for fruits and vegetables [81,82,83].

While most of the literature present tests of antibacterial activity on strains that usually spoil the food (*Escherichia coli, Staphylococcus aureus, Salmonella enterica,* etc.) [84], there are some authors who test the packaging materials also against various fungal strains (yeasts and molds) [85]. Developing new packaging materials must pass some specific tests beside the above-mentioned antibacterial activity. Some additives (usually natural extracts or specific substances) have antioxidant activity that aids in prolonging shelf life. The new packaging material must have good mechanical resistance and barrier properties. The water vapor permeability (WVP) plays a crucial role in preserving the food. Permeability to other gases, namely oxygen, carbon dioxide, ethylene, and various aromatics is also very important when polymeric film is designed. Some other characteristics like transparency, thermo sealability, leakage of additives etc., will be described further.

The problem of food loss due to spoilage by microorganisms can be addressed by using antimicrobial packaging, be it traditional fossil-based polymers (PVC, PET, PE, PP, etc.) or biodegradable ones (cellulose, starch, chitosan, etc.). The ecological impact generated by the plastics cannot be solved unless we replace the traditional packaging materials with biopolymers from renewable sources.

The aim of the review is to present some of the latest research trends in the antimicrobial food packaging, the work horse polymers, antibacterial and antifungal agents, as well as to review the strategies presented by the literature. We searched for original articles and reviews that addressed the antimicrobial packaging subject. In order to identify the strategies and latest trends in the domain of antimicrobial food packaging we survey the Scopus, Clarivate, and Science Direct databases with emphasis on 2017–2020 interval. The search engines were used to find relevant documents by using keywords as “antimicrobial food packaging” “antimicrobial peptides” “biopolymers” “essential oils” and “antimicrobial peptides” Additional documents were identified by examining references listed in key articles. The initial number of documents obtained by searching the databases was 275,125. By restricting the time interval, the number dropped to 88,327. Based on title and abstract information we included only the articles that were considered relevant for antimicrobial food packaging subject. Therefore, 289 articles were included in this systematic review.

## 2. Antimicrobial Packaging Obtained by Modification of Current Materials

Nowadays there are two main research directions for the packaging industry. First research approach is to modify the current packaging materials in order to confer antimicrobial and antioxidant activities, plus better mechanical and barrier properties. This direction is somehow closer to the manufacture sector, which will not be forced to make radical changes into the production lines, so the implementation costs are lower. While the consumer will enjoy a safer food with longer shelf life, the generated pollution is the main drawback of this approach. The nonbiodegradable plastics will still be produced and will still be an environmental hazard. The second approach, with increasing support from consumers and researchers is to replace all the nonbiodegradable polymers with biodegradable ones, obtained from renewable sources. This is more ecological but is also more costly, demanding a dramatic change of the packaging manufacture sector.

For large quantities from some foods (sugar, flour, rice, potato, beans, etc.) the storage and transport can be done in textile bags. Therefore, some researchers aim to obtain fabrics with antibacterial and antifungal properties (that can also be used outside of food industry, for clothing for example). One method presented in the literature is to load the fabric with quaternary ammonium salts that contain isocyanate moiety (CAI) and sulfopropyl betaine (SPB) by soaking-drying process. Both moieties, SPB and CAI, can be covalently bounded to the fabric surface and will give excellent bactericidal and antifungal properties. The effect is long-lasting and cannot be washed away (Figure 2). This functionalization of cotton fabric will not modify significatively the permeability to water vapor or air or its hydrophilic character. The mechanical properties of the fabric are enhanced after the treatment, the textile bag presents better tearing and breaking properties. Increasing the CAI load leads to a stronger covalent bond and the ratio SPB/CAI can be used to enhance the retention of the antibacterial agents. The fabric has not exhibited any noticeable migration of the chemical agents, the test indicating a strong retention into the cotton fiber. As a consequence there is no toxicity of stored food due to the migration of CAI or SPB [86].

By same general method the antimicrobial activity can be obtained for other innovative packages. The deposition or the embedding of bactericidal and antimycotic agents into the polyvinyl chloride (PVC) [87], polyethylene terephthalate (PET) [88], polyethylene (PE) [89], or polypropylene (PP) [90] films that are used normally for food packaging, do not request a fundamental change of the production lines and the new types of packaging can easily enter production.

Mixed cellulose/PP pillow packages can be used to extend the shelf life of iceberg lettuce if emitting sachets with eugenol, carvacrol, or trans-anethole are put inside [91]. The sachets will slowly release the natural antimicrobial agent and help preserve the food. This is an easy, low-cost method to confer antibacterial activity to the packaging materials already in the market and can be classified as modified atmosphere packaging (MAP), and can be seen as an intermediate step toward obtaining antimicrobial packaging without addressing the problem of plastics pollution [92].

Small modifications for actual packaging materials are described in [93], where the authors have obtained a bilayer film from low-density polyethylene (LDPE), in which one layer has incorporated various essential oils. PVC was also modified by adding orange essential oil and the obtained films were proved to have antimicrobial activity [94]. These traditional polymeric films can have silver nanoparticles (AgNPs) or oxide nanoparticles embedded into them or deposited on the surface. The polymeric film or the nanoparticles can in turn be loaded with essential oils as antimicrobial agents. PVC film with embedded nanoparticles like AgNPs [87], TiO_2_ [95], ZnO [96] are reported in literature.

The AgNPs can be loaded on silica, and these in turn can be embedded into the PP film. This kind of composite film can be obtained by hot pressing the silica powder onto the PP film. The antimicrobial film thus obtained has superior mechanical properties and the test made on fresh mackerel indicate a longer shelf life. In the samples with AgNPs/SiO_2_ 5% and 10% respectively concentrations of 3.06 and 3.11 log CFU/g for pathogen agents were recorded, while into the control sample the concentration reached a peak of 4.81 log CFU/g. In the same time the level of trimethyl amine increased from 1.6 to 2.5 mg/100 g in the control sample, but there was no modification in the samples packaged with composite films. The color of the fish meat stays red for samples with composite film, but changed from red to yellow for the control sample [97].

Polyethylene terephthalate (PET) can be used as antimicrobial packaging after surface modification. Because of its inertness the PET film requires an activation treatment so that functionalization/grafting can succeed. Antimicrobial peptides (AMPs) are an important class of amphiphilic molecules capable to suppress growth of microbes. They are composed of 12 to 100 amino acids and therefore are quite different structurally. Functionalization of PET with an AMP, mitochondrial-targeted peptide 1 (MTP1), is presented in by Gogliettino, M. et al. [98]. The antimicrobial packaging PET-MTP1 was used to prolong the shelf life from 4 to 10 days for meat and cheese.

Among AMPs, nisin and pediocin are used commercially, but other species such lacticin 3147 or enterocin AS-48 have been proposed for industrial applications. The prime target are the dairy products as these bacteriocins can be produced by lactic acid bacteria [99]. The main problem is to find ways to immobilize AMPs on film’s surface without losing the antimicrobial activity. Direct incorporation, coating, or immobilization are the main strategies [100]. Various methods are proposed in literature, such as cold plasma treatment [101], enzyme (laccase) surface modification [102], adsorption on nanoparticles [103] etc.

Among the desirable properties that a film must possess to be used in food packaging is the heat-sealing capability. While traditional PE or PP films can be easily heat sealed, some of the biopolymers, like cellulosic films made from nanofibers, lack sealability. This problem has made the industry reluctant to market them. This shortcoming can be overcome by using the traditional PE or PP films on which cellulosic nanofibers can be deposited. This composite film will present the advantages from both materials. Peng Lu et al. [104] have demonstrated that these cellulose nanofibers can also be loaded with various antimicrobial agents, like nisin for example, to enhance the film properties (Figure 3). PE and PP films have been treated with cold plasma in order to improve hydrophilic character and to make them compatible with cellulose fibers. The cellulose layer has led to a decrease of the oxygen permeability from 67.03 to 24.02 cc/m^2^·day. Adding the nisin (an AMP) has no influence on permeability, but has conferred antibacterial properties to the films. The test made against *Listeria monocytogenes* have showed a kill ratio of 94%, indicating the economic potential of this kind of packaging.

Cellulose-based materials play an important role in the packaging industry, therefore, many researchers are trying to improve them. The poor mechanical performance and the lack of water resistance are some of the factors that limit the applicability domains. Zhu R. et al. [105]. have proved that by applying recycled cellulose, obtained in ionic liquids, over plain paper, the mechanical properties are improved greatly. Moreover, by using only 2% cellulose in paper composition the oxygen permeability drops by a factor of 10^6^. Although the composite is hydrophilic in nature, it has good resistance to water action and keeps its form when submerged.

Traditionally, for improving the WVP values, the oil resistance, or the mechanical properties of paper packaging, at least one side is usually covered by a thin film of PE, or PET to a lesser extent. Therefore, in the quest of replacing the fossil-based polymers with biodegradable ones, researchers come with innovative composite materials that present new and superior properties. A method of enhancing the properties of paper packaging is described in [106] where the plain sheet was covered with a functionalized biopolymer coating. The film was obtained by mixing alginate with carboxymethyl cellulose and carrageenan, and was loaded with grapefruit seeds extract. The microscopy studies indicate that the biopolymer is compatible with the paper support, filling the pores and levelling the spaces between the cellulose fibers, in the end resulting a smooth surface. The whole range of properties for this type of packaging: water and oil resistance, WVP, hydrophobicity, and mechanical properties were not only superior to the simple paper support but also to paper-PE packages used at present. The obtained biopolymer film presented antibacterial properties against two pathogens *L. monocytogenes* and *E. coli*, which were destroyed in 3 and respectively 9 h of exposing. The test made on fish meat inoculated with bacteria have demonstrated the total elimination of the infection in 6–9 days, proving the economic potential of this packaging type [106].

A similar strategy for improving paper packaging is described in [107] where cellulosic paper was covered with a chitosan layer by dip-coating method. The chitosan solution also contained nisin up to 1 g/mL as antibacterial agent for the packaging material. Various compositions based also on chitosan with the addition of zein and rosemary oil coated on paper were tested by Brodnjak and Tihole [108]. As expected, the WVP values dropped sharply after coating the paper, while the new packaging exhibited antimicrobial activity.

Such experiments have demonstrated the possibility of obtaining antimicrobial packaging, usually as a nanocomposite film, based on materials that are currently in use in the food industry. The increase in production cost is one of the main drawbacks for the introduction of new technologies. The industry will adopt new technologies that can be re-engineered from the existing ones with minimal modification of the production lines. These modifications can be induced by new regulations, by predicting increasing profits or by changes in public perception and demand [109].

## 3. Biodegradable Polymeric Antimicrobial Packaging

Since their invention in 1959, a great number of plastic bags and other plastic packages have been produced from fossil fuels and their contribution to the pollution has become a global problem. The obvious solution is to produce bags and packages based on biodegradable materials, with low environmental impact. By adding antimicrobial properties to the biodegradable packages these new materials can offer an enhanced protection against food spoilage, extending the shelf life.

In the search of alternative materials to the plastics, the scientists have put a special emphasis on biopolymers. One of the most important conditions, the availability, is fulfilled by some abundant natural polymers like cellulose (and derivatives like carboxymethyl cellulose—CMC, methyl cellulose—MC, cellulose acetate—CA, etc.), chitosan, lignin, starch, polylactic acid (PLA), polyvinyl alcohol (PVA), alginic acid and derivatives etc. (Table 1). To be used in the food industry, these materials must be nontoxic, renewable, and present the right properties. Although many of them are included in the polysaccharides class and possess a similar monomeric structure (Figure 4), the polymeric chain configuration, hydrogen bonds, and substituents play a crucial role in shaping the specific properties of each biopolymer. The biopolymer films must be resistant, to not exfoliate and to not permit the gas or vapor exchange between food and atmosphere. Some of the most abundant biopolymers have also antimicrobial intrinsic activity (like chitosan), but usually in the native form they have low mechanical resistance and poor barrier properties. Unfortunately, cellulose, the most abundant biopolymer, lacks any antibacterial activity and therefore it must be loaded with antimicrobial agents. These agents can also become reinforcing components improving the mechanical properties and enhancing the WVP values.

PVA has high hydrophilic properties, is water soluble, with a good crystallinity, and has many hydroxyl moieties at side chains which can form hydrogen or covalent bonds with other polymers or can help loading various antimicrobial agents [145]. PLA can present good crystallinity, but is soluble only in organic solvents, and therefore reinforcing it with nanoparticles requires surface compatibilization [146]. Alginic acid and its sodium salt are easily water soluble to be used raw. Therefore they are mixed with lipidic compounds (like essential oils which also give antibacterial activity to the final material) and reinforced with various nanoparticles [147]. Chitosan is water soluble (acidic pH) and can also provide -OH and -NH_2_ moieties for hydrogen bonds. Cellulose and its derivatives represent a class of hydrophilic biopolymers, which again present free hydroxyl moieties. The strategy for improving the water resistance of biopolymers with -OH moieties is to mix them in composite materials, where same moieties will help them cross-linking. This blending of individual polymers gives many hydrogen bonds which improve the mechanical properties of the composite. Sometimes the simple blending of two polymers does not yield automatically a better material. PVA/CMC composite films have low water resistance and present no antimicrobial activity for example. Adding inorganic nanoparticles (like Ag, ZnO, TiO_2_, Fe_3_O_4_, etc.) and essential oils in such polymeric films will improve both WVP and antimicrobial activity [111,148,149,150,151].

In the food industry, the water vapor and oxygen barrier properties of the packaging are crucial, but the permeability to CO_2_ and aromatic compounds plays an important role too. The engineering of the composite films generally aims to enhance the properties of the base biopolymer (usually a polysaccharide), in order to obtain the desired properties. Also, in the active or intelligent packaging the nanomaterials can act as antimicrobial agents, oxygen scavengers, or as signals for improper storage conditions. Compatibilization between nanomaterials and the polymeric matrix is the main challenge in obtaining antimicrobial packaging [152].

The bio-nanocomposite materials represent an important alternative to the traditional fossil-based plastics used as packaging. These are biocompatible, biodegradable, and can exhibit superior performances, both mechanical and chemical. Biopolymers like chitosan, carboxymethyl cellulose, starch, alginate, casein, carrageenan, or cellophane can solve the ecological problem of packages due to their biodegradability and non-toxic nature. But polysaccharides also have some drawbacks beside their clear advantages. Some mechanical properties are inferior, and they have a low water resistance or high permeability. Consequently, by embedding various nanomaterials into the natural polymeric film better thermal and mechanical properties can be obtained, but also the water vapor and oxygen barrier properties can be enhanced, without losing the biodegradability or the non-toxic character of the biopolymer. The most used are clay nanoparticles (like montmorillonites, kaolinite, or laponite), oxide nanoparticles (ZnO or TiO_2_), or silver nanoparticles (AgNPs). In Table 2 are presented some types of antimicrobial packaging materials, their main polymer, antimicrobial agent, and the tested kind of food.

### 3.1. Edible Films

The edible films represent a protection packaging, with antimicrobial activity for the food safety. The literature reports a large variety of materials which can form films for food packaging, ultimately their properties indicating the exact food on which the film or coating can be used [180]. Edible packaging materials or coatings must be non-toxic, adherent to food surface, be tasteless or present an agreeable taste, have good barrier properties and prevent water depletion of the food, must have a good stability in time and prevent the mold formation, good appearance in order to be accepted by consumers, and must be economically viable.

Edible films are composed of polysaccharides, proteins, or lipids and they must present better barrier properties and antimicrobial activity in order to extend the shelf life. Among polysaccharides we can enumerate starch, cellulose derivatives, alginate, chitosan, pectin, carrageenan etc. Proteins can be extracted from animal sources like casein, whey, collagen, gelatin, egg white, etc. or can be from vegetal sources like corn zein, gluten, soy proteins, rice bran, peanut, keratin etc. The third category, lipids, is composed of saturated or unsaturated fatty acids.

The use of edible films for food packaging is not a new technology. In the past collagen-based membranes were used for various meat products like sausages, hotdogs, salami etc. At present, the food industry is searching for alternatives that can also prolongate the shelf life of the products [181].

The edible films for fruits, cheese, and meat products are usually based on natural, abundant, cheap, edible polymers, which can effectively enhance the quality of the food and at the same time can reduce the quantity of foodstuff that is altered. This kind of films can replace successfully the waxy coatings that are used on various fruits (apples, oranges, limes etc.) and that are toxic and are forbidden by an increasing number of countries. Edible biopolymers inhibit the food alteration and complies to the directives requested by EU market: organic food, healthier, and with longer shelf life.

### 3.2. Polysaccharides-Based Films

Chitosan, one of the most abundant natural polymers, is the work horse for such innovative edible antimicrobial packaging. It is a polysaccharide, with intrinsic antibacterial activity, usually extracted from shells of crabs and shrimps. It also has some antioxidant properties and is biocompatible, biodegradable, and most of all is edible.

A chitosan packaging film can protect the food by multiple mechanisms. The film can block the microorganism’s access to the food as any physical barrier. It can hinder the respiratory activity by blocking oxygen transfer and can cause a physical obstruction so that microorganisms cannot reach the nutrients (mechanisms I and II—Figure 5). Various nutrients can be chelated by chitosan chains (mechanism III) and the outer cellular membrane of the microorganisms can be disrupted by electrostatic interactions (mechanism IV), all these adding a supplementary stress which microorganisms must endure. The antibacterial activity of chitosan can also be the result of damaged cell membrane, which will lead to leakage of intracellular electrolytes (mechanism V). Once the chitosan chains have diffused inside through cellular wall, there are multiple action pathways that leads to microorganism’s death. It can chelate internal nutrients or essential metal ions from cellular plasma (mechanism VI), it can influence the gene expression (mechanism VII) or it can penetrate the nucleus and bind the DNA, thus inhibiting the replication process (mechanism VIII) [182].

Among its drawbacks are the higher cost, the acidic hydrolysis and weak mechanical properties. Therefore, in many cases it will not be used alone but in a nanocomposite film, in which the other components (polymers or nanoparticles) will compensate for the poor performances of chitosan in some areas. Usually improving the mechanical properties of polymers requires some plasticizers. Chitosan has a good compatibility with glycerol, xylitol, sorbitol etc.

The use of chitosan in edible films can raise the problem of its origin. Traditionally the chitosan is obtained from the marine crustacean’s exoskeleton, which would make it undesirable for strict vegetarians or for halal and kosher food. Nevertheless, it can be obtained also from alternative sources like mushroom inferior stem (which itself is a residual product that is usually discarded) [183].

Priyadarshi, R. et al. [184] have obtained chitosan-based antimicrobial packaging by forming the polymeric film directly on the surface of the food. As binding agent, they used citric acid, that also enhanced the film stability and antioxidant properties. As plasticizer they used glycerol, which gave flexibility to the film. In fact, the elongation capacity was increased 12 folds, but this decreased the tear resistance. The as-obtained packaging material had a superior water resistance, reducing WVP by 29%. Also, the films were transparent, which is desirable from consumers’ point of view, as they will always want to be able to see the food inside the packaging. Finally, the test made on green chili indicated an extended shelf life [184].

Enhancement of chitosan film properties was done by adding layers of other polymers and embedding nanoparticles. Bilayer films obtained from chitosan and polycaprolactone (PCL) have been obtained by pressing or coating. Both layers have been loaded with nanocellulose (2–5%) and grape seeds extract (15% *w/w*). The presence of nanocellulose (NC) have reduced significantly the WVP and the film opacity, while the grape seeds extract had the opposite action. The films obtained by pressing exhibited higher values for elastic modulus and stretch resistance than the films obtained by coating from solution. Both film types had antimicrobial activity and the grape seeds extract preserved the antioxidant activity while being loaded into the chitosan matrix [185]. Similar mechanical behavior was reported in [186] where authors obtained a composite chitosan-based film by adding cellulose nanocrystals and grape pomace extracts. Adding NC increased tensile strength and decreased elongation and WVP while the grape pomace extract had opposite effect.

Uranga J. et al. have created antibacterial films from chitosan and citric acid, but also added fish gelatin into the composition. The films have antibacterial activity on *E. coli*, act as a UV barrier, have good mechanical properties, and the citric acid acts like an inhibitor against swelling in the presence of water [187]. Such films can be used for products like seafood. In the case of fish or shellfish the food alteration happens because of the modification induced by enzymes activity that are naturally presented in meat, reactions like lipids oxidation or because of the microorganism’s metabolism. Chitosan films can reduce the lipids oxidation reactions, although the native antioxidant properties are not so good. To overcome this problem, Uranga, J. et al. developed a new strategy by adding into the chitosan—fish gelatin mix, anthocyanins obtained from processing food wastes. Anthocyanins preserved antioxidant activity during the mold pressing process, the obtained packaging exhibiting antimicrobial and antioxidant activity, along good mechanical and WVP properties [188]. Other groups loaded the chitosan-gelatin film with procyanidin or various other antioxidants [189,190]. Enhancement of antioxidant activity of chitosan films can be done also by adding some radicals like 2,2,6,6-tetra-methylpyperidine-1-oxyl. The desired mechanical properties can be achieved by adding up to 15% cellulose nanofibers and a plasticizer like sorbitol (25%). Beside the antibacterial effect on *E. coli,* the films also reduced the proliferation for *Salmonella enterica* or *L. monocytogenes* [191].

Another type of edible chitosan-based film was obtained by Jiwei Yan et al. [155] and used for fresh strawberries. The fruits were covered with a bicomponent film (layer-by-layer LBL) by successive immersions in solutions of 1% chitosan and 1.5% carboxymethyl cellulose (Figure 6).

The results have indicated a significant enhancement of the LBL film properties when compared with the simple chitosan one. The bicomponent film has conserved the fruits for longer time, preserving the consistency, the flavor and allowed only minimal changes in acidity and total soluble substance because the LBL film has decreased the metabolic degradation of carbohydrates, fatty acids, and amino acids from the fruits, preserving them for eight days [155].

The thymol, a component from thyme essential oil, has also been used as antibacterial agent in the edible chitosan films. A nano-emulsion of thymol was loaded onto a film made from chitosan and quinoa proteins. Strawberries covered with such films have presented a superior resistance against fungi. If initially the edible film has somehow altered the fruits flavor, starting from the 5th day the sensorial qualities have been enhanced when compared with control batch. In addition, the shelf life was extended by four days, mass loss was diminished, and parameters like pH, titratable acidity or soluble substances percent were not altered [192].

Another composite film that has attracted many researchers is the chitosan/ polyvinyl alcohol (PVA). This composite presents good biocompatibility and low toxicity, has broad spectrum antibacterial activity on both Gram-positive and Gram-negative strains, exhibits good thermal stability, and can be loaded with desired antimicrobial agents. Yun, Y.H. et al. [193] have prepared mixed polymeric films based on chitosan and PVA for fruit preservation. The films with 1:1 ratio between polymers, used sulfosuccinic acid as the binding agent, therefore the reticulation could be made by UV light exposure (approx. 20 min). Glycerol, xylitol, and sorbitol were used as plasticizers. The composite with sorbitol presented the best mechanical and thermal properties. All the samples had lower WVP than simple polymeric films. The reported results indicate that after 70 days storage, the fruits presented negligible decay. The packaging material is biodegradable, test showing a 40–65% disintegration after 220 days. Yang, W. et al. [194] have obtained a hydrogel by mixing chitosan and PVA in which they added 1–3% lignin nanoparticles. Lignin is the second most abundant natural polymer after cellulose, but usually is used in low value products, mostly for burning. In this hydrogel, lignin exhibited antioxidant activity and UV filter properties, while being also a reinforcing and antibacterial agent (Figure 7).

Best properties have been obtained for 1% lignin as higher quantity gave no further benefits because of agglomeration of nanoparticles. The hydrogel has a porous structure like honeycomb, with micron size pores which can be further loaded with antimicrobial agents. The tests made by the authors indicate a synergic activity between chitosan and lignin against *E. coli* and *S. aureus*, and also enhancing of antioxidant properties [194]. Youssef, A.M. et al. have improved same polymeric mix chitosan/PVA by using glycerol as plasticizer and adding TiO_2_ nanoparticles. The coating was successfully used to preserve Karish cheese for 25 days, best results being obtained for 3% TiO_2_ [174].

The chitosan-based films can also be used on fruits and vegetables that are ripening under ethylene action. Because such films can absorb the ethylene, they can postpone the unwanted ripening. The literature mentions the composition 6 g chitosan: 7 g KMnO_4_: 2 mL sorbitol, to preserve tomatoes up to five times longer at room temperature [195]. Similar findings have been reported for other nanocomposite chitosan-based films. In order to enhance the antibacterial properties of the chitosan coating, TiO_2_ nanoparticles can be embedded into the film. The coating can be applied straight from the solvent on the packaging. Such films have been used to protect the cherry tomatoes and to postpone the ripening after harvest. The authors have monitored the consistency, color changes, lycopene quantity, mass loss, total soluble substance, ascorbic acid, and concentration for CO_2_ and ethylene inside the package. The conclusion was that the addition of the TiO_2_ nanoparticles have enhanced the packaging quality, the cherry tomatoes presenting less changes when compared with simple chitosan packaging or with the control lot. Authors have attributed the behavior to the photocatalytic activity of TiO_2_ versus ethylene gas (Figure 8) [154].

Chitosan is compatible also with ZnO nanoparticles, the composite having a more potent antibacterial activity due to synergism [54]. Good results are reported for example by [196] who manage to reduce the proliferation of *E. coli* and *S. aureus* on poultry by 4.3 and 5.32 units on log CFU/mL when compared with control. Generally speaking, the meat products have a high nutritional value and represent a good medium for microorganisms growth, thus being preserved only for a few days in the fridge. Therefore, it is no surprise that many researchers have obtained chitosan-based packaging for meat products [197,198].

A chitosan / cassava starch film was obtained by Zhao, Y.J. et al. [172] by using subcritical water technology. The authors added glycerol as the plasticizer and gallic acid in order to achieve antimicrobial and antioxidant activity for the film. The formation of new hydrogen bonds, electrostatic interactions, and ester bonds between starch, chitosan, and gallic acid had the effect of decreasing permeability to water vapors. Antimicrobial tests on foodborne diseases microorganisms have indicated a prolonged shelf life for pre-inoculated ham from 7 to 25 days when compared with the control. Various edible films based on chitosan and starch (extracted from oak) have been investigated by Zheng K. et al., who have indicated that increasing the starch ration from 0 to 50% has improved the mechanical properties by 20%, has decreased the WVP by 3%, and diminished the oxygen permeability by 2.5 times. Aiming to improve the antimicrobial activity, the authors have added essential oil from *Litsea cubeba*, which further increased the barrier capabilities of the packaging [199].

As mentioned before, by incorporating essential oils into chitosan films the antimicrobial and antioxidant activity is usually greatly enhanced. The literature presents many reports of such films that are efficient against food altering pathogens, bacteria, and fungus, that spoil the harvest. The use of essential oils and plant extracts with edible natural biopolymers is one of the newest research trends in food packaging industry.

Some essential oils are produced from crops that are especially made for this purpose (lavender, thyme, cinnamon, basil, rosemary, etc.). Other essential oils or extracts are obtained as by-products processing agricultural wastes (grape seeds, apricot kernels, orange peel, etc.). One of the challenges in this type of food packaging is that the producer must match the type of essential oil with the type of the protected food. While thyme, ginger, or rosemary essential oil will be well received by consumers on the meat-based products, not many consumers will like them with fruits. Same can be said about cinnamon and vanilla which are compatible with pastry, but do not go so well with meat. In the same time the essential oils from orange or lime can be used in fruits or seafood packaging (Table 3).

Apricot kernels are a major agricultural waste. One possible utilization for them is to extract the essential oil from the bitter core. The chemical composition for this essential oil indicates the oleic acid as the main fatty acid, but also reveals the presence of *N*-methyl-2-pyrolidone which is a strong antioxidant and antimicrobial agent. By mixing chitosan with apricot essential oil in 1:1 ratio, the WVP dropped by 41% and the mechanical resistance increased by 94%. The antimicrobial activity was tested on bread slices, on which the chitosan-essential oil package managed to block the fungus growth [206]. The possible use of agricultural waste was further exemplified by Zegarra, M. et al. who have used chitosan films loaded with acerola extract from residues to extend the shelf life of meat [164].

Essential oil from *Perilla frutescens (L.)* Britt. was introduced by Zhang Z.J. et al. in a chitosan-based film, ranging from 0.2–1% *v/v*, in order to enhance the antibacterial properties against *E. coli, S. aureus*, and *Bacillus subtilis*. The essential oil also enhanced the mechanical properties, the UV blocking capacity and decreased the WVP and swelling capacity [207].

Another essential oil with good antioxidant and antimicrobial activity is extracted from cinnamon. It improves the antibacterial activity of the chitosan-based film, but also decreases the oxidation speed for lipids and reduces the WVP. Tests by Ojagh S.M. et al. have shown that if after 16 days the control sample had a bacterial load of 8.43 log CFU/g, the chitosan presented a bacterial load of 6.79 and the sample packed in chitosan-essential oil had a bacterial load of only 6.68 log CFU/g (for both chitosan-based films the bacterial load being under the limit of 7 log CFU/g) [208]. Same combination of chitosan–cinnamon essential oil has been tested for increasing the preservation time of jujube (date) fruits. Normally, these fruits have a shelf life of only 15 days and the orchards cannot be efficiently exploited as the post-harvest losses are too high. The use of fungicides is not acceptable because of health concerns and any way the ripe fruits have the highest decay rate. Xing Y. et al. have shown that chitosan–cinnamon essential oil films have lower WVP and therefore prevent the mass loss. The test indicates that the shelf life can be increased to 60 days as the decay rate is drastically reduced. The antimicrobial activity was tested on *E. coli*, *S. aureus, Rhizopus nigricans*, *Penicillium citrinum*, *Aspergillus flavus,* and *Penicillium expansum,* the best results being observed on *R. nigricans* and *S. aureus* strains [209].

Chitosan films have a poor performance regarding WVP, and this drawback must be removed by structural modifications. The solution is to add some lipidic compounds which are hydrophobic. Best candidates are fatty acids and essential oils or just compounds from essential oils. The use of selected compounds is a strategy to remove some of the unpleasant or too strong flavor of essential oil, or to avoid including potential toxic components into the packaging. Wang, Q. et al. have loaded the chitosan film with carvacrol and caprylic acid. Carvacrol is one of the major components of oregano and thyme essential oils with good antimicrobial and antioxidant activities and is listed as GRAS (generally recognized as safe) by FDA. The caprylic acid is also listed as GRAS, has a broad antibacterial spectrum and is naturally found in palm seed oil, coconut oil, or ruminant milk [162]. The edible films were used as packaging solution for white shrimps, *Litopenaeus vannamei*, for 10 days. During this time the growth of microorganisms was hindered, nitrogen volatile compounds were not evolved, and the pH variation was small when compared with control sample. The surface melanosis was delayed, and the organoleptic properties of the shrimps were enhanced. Research indicates that combining the chitosan with carvacrol generated a synergic effect, and adding the caprylic acid further enhanced the antimicrobial activity of the film [162]. Pabast, M. et al. have obtained better WVP properties for chitosan-based films by adding the *Satureja khuzestanica* essential oil encapsulated into nanoliposomes. The nanoliposomes of ~95 nm were obtained from a mix of soy-lecithin and had an encapsulation efficiency of 46–69%. By encapsulating the essential oil into nanoliposomes the release rate has decreased, which led to a prolonged antibacterial and antioxidant activity. The meat packed with this film was preserved for 20 days at 4 °C without spoilage [170].

Cellulose is the other most common polysaccharide used for edible films and for biodegradable packaging materials. It can be obtained from a great variety of sources (vegetal or bacterial). The poor mechanical performance and the lack of water resistance are the factors that limit the applicability domains. By loading the cellulose with an antimicrobial agent, the added value to packaging materials increases. The literature presents many cellulose-based edible films, for example obtained from CMC and glycerol mixed with essential oil from lemon, orange, or other citrus as antibacterial agent. The tests have indicated substantial increases of the antibacterial activity against *E. coli* or *S. aureus* by adding up to 2% essential oil (MIC 100, 250, and 225 mg/mL for lime, lemon, and orange) [210].

Ortiz, C.M. et al. have used microfibrillated cellulose mixed with soy proteins and glycerin as plasticizer to obtain biodegradable packaging. Further, the film was loaded with clove essential oil to enhance the antimicrobial and antioxidant activity against spoilage microorganisms. The addition of the essential oil had also the effect of a plasticizer, improved WVP, but increased oxygen permeability. The cellulose fibers had 50–60 nm diameter and ~35% crystallinity [211]. A similar composition based on soy protein, cellulose nanocrystals, and pine needle extract was reported by Yu, Z. et al. [212]. The use of cellulose nanocrystals and pine extract lowered WVP for the film by impeding on hydrogen bond formation and increased the mechanical strength.

In order to improve the mechanical and barrier properties cellulose can be reinforced with other polymers or with nanoparticles. The normal choice rests with natural, abundant polymers, which can be used as they are or after chemical modification [213]. Cellulose can be combined not only with other polysaccharides [214] but also with proteins [215,216] or lipids [217,218]. Zhang X. et al. have obtained by coagulation into a LiCl/ dimethyl acetamide/AgNO_3_/PVP solution, in one step synthesis, a cellulosic film with embedded AgCl crystals. An important role for the embedding process is played by the presence of PVP. By light exposure some of the AgCl was decomposed and Ag@AgCl nanoparticles were generated. The shape and size of these nanoparticles can be tuned by PVP concentration. The test made on *E. coli* and *Staphylococcus aureus* have demonstrated a strong antibacterial activity, after three hours of exposure no viable bacteria being identified [219].

Zahedi, Y. et al. reports the synthesis of a composite film based on CMC with 5% montmorillonite and 1–4% ZnO [220]. The addition of these nanoparticles decreased WVP values by 53% and enhanced the mechanical resistance. The nanoparticles also were an efficient UV blocking agent. Tyagi, P. et al. obtained an improved paper-based package material by coating it with a composite made of nanocellulose, montmorillonite, soy protein, and alkyl ketene dimer [221]. When researchers obtain similar composites based on cellulose or chitosan (with montmorillonite and essential oils for example) the intrinsic antibacterial activity of chitosan will always make the difference between them [222]. Elsewhere [223], AgNPs immobilized on laponite were obtained with quaternized chitosan and used for litchi packaging. Turning chitosan into a quaternary ammonium salt or subjecting it to other chemical modifications is an efficient way to enhance its properties. CMC can be mixed in any proportion with such modified chitosan as the two substances are miscible because of the formation of hydrogen bonds. The composite film has better mechanical and thermal performances. The WVP is decreased, but the oxygen permeability is increased in such composites. The CMC-chitosan composite has antibacterial activity against *S. aureus* and *E. coli*, and the test made on fresh bananas indicated an increased shelf life for the coated fruits [224].

Bacterial cellulose (BC) from *Gluconacetobacter xylinus* can be transformed into cellulose nanocrystals by acidic hydrolysis. These BC nanocrystals with size of 20–30 nm, together with silver nanoparticles (AgNPs) with size of 35–50 nm can be embedded into chitosan films. The embedding process has a significative influence on the color and transparency of the films, but it greatly enhances the mechanical and barrier properties. The forming of new hydrogen bonds between chitosan and BC indicate that the embedding is not only physical, but also implying a chemical process. The antimicrobial properties of these films vs. food specific pathogenic agents, suggest a synergic action of chitosan and BC-incorporated nanoparticles [225]. Incorporation of AgNPs into the methylcellulose films can be achieved with *Lippia alba* extract. The films have a smaller mechanical resistance and elastic modulus than the control lot, but they are easier to stretch and have a higher hydrophobicity. In the same time the antioxidant and antimicrobial activity are higher when compared with same control films [226].

The literature reports also nanocomposite films based on cellulose that contain not only AgNPs but also oxide nanoparticles from transition metals like CuO or ZnO. During production of regenerated cellulose fibers from cotton or microcrystalline cellulose from BC, oxide nanoparticles can be attached to the fiber surface by multiple hydrogen bonds (generated by hydroxyl moieties), without any additional functionalization. The number of oxide nanoparticles attached to the cellulose is nevertheless smaller than the AgNPs that can be loaded. The thermal stability of the composite films with AgNPs and ZnONPs is higher if compared with bare cellulose film. Antibacterial tests done on *E. coli* and *L. monocytogenes* indicate a strong inhibition of this strains [227].

Starch is also a common polysaccharide used for edible films, and unlike cellulose, it has also a nutritional value for humans. Simple starch films mixed with various natural compounds can form efficient antimicrobial packaging materials [228]. Starch films loaded with clove leaf oil can preserve *Listeria* inoculated cheese for 24 days. The clove leaf oil fulfills multiple roles, as it increases tensile strength and elongation break, but also inhibited *L. monocytogenes* proliferation, acted as UV barrier and radical scavenger [176]. An innovative design based on cheese whey and starch from agricultural waste is presented by Dinika et al. [229]. In this case the AMP from cheese whey are responsible for the activity of the final edible antimicrobial packaging film. Trongchuen, K. et al., have also obtained a starch-based composite that can be used as antimicrobial packaging material. Tests indicated a good activity against *E. coli* and *S. aureus* for composition containing starch foam, 10% extracted spent coffee ground (SCG), and 8% oregano essential oil [230]. A similar composition but with enhanced antimicrobial and antioxidant properties was developed by Ounkaew A. et al. who mixed starch with PVA and in the resulting nanocomposite-incorporated SCG and citric acid [231].

Other authors used bentonite clay to reinforce the starch-based film (cassava starch) in order to improve the mechanical properties. As plasticizer 2% glycerol was used. To ensure antimicrobial activity, 2.5% cinnamon essential oil was added. Antibacterial tests on *E. coli*, *S. aureus,* and *Salmonella typhimurium* confirmed the capacity of this packaging material to protect the food against such strains. Test made on meat balls indicates that the film was able to extend the shelf life up to 96 h (keeping the bacterial population under the FDA recommended value of 10^6^ CFU/g) [232].

### 3.3. Other Biopolymers-Based Films

Another biopolymer that has attracted much attention is the PLA, which can be easily produced by bacterial fermentation of some renewable resources like corn starch or sugar beet. At the end of economic life, the PLA products are easily decomposed as they are biodegradable. The main problem for PLA films to be accepted as a good candidate for food packages is the poor barrier properties, as oxygen can easily permeate thru PLA films.

Composite based on PLA and nanocellulose loaded with plant essential oils (from *Mentha piperita* or *Bunium percicum*) were used to extend shelf life for grounded beef up to 12 days in refrigerated condition [171]. While the authors report no significant differences of WVP when adding nanocellulose to PLA, the values decrease to 50% when essential oils are added. Therefore some authors studied the possibility of using PLA films loaded only with some major components of essential oils as antimicrobials [233]. Better properties are reported when PLA is mixed with proteins before being loaded with essential oils [234].

Another way to improve the WVP is presented by Zhu J.Y. et al., who have obtained bi and tri-layer films from PLA and gliadin, without surface modification [158]. These films have a good mechanical resistance, do not exfoliate, and most important, the gliadin improves the barrier properties of the PLA toward oxygen, while the PLA improves the barrier properties of gliadin toward water vapor. By loading the films with antimicrobial agents such thymol, the authors have obtained a biodegradable polymeric film with antibacterial activity, which is suitable for food packaging. Moreover, the film has exhibit different antimicrobial activity on each side, the gliadin side being more potent than the PLA, indicating a preferential release (Figure 9).

Yin, X. et al. have also synthetized PLA films in which they added salicylic acid (SA). SA is heavily utilized after harvesting for fruits and vegetables preservation. The authors have demonstrated that if PLA film is loaded with up to 6% SA the antibacterial activity against *E. coli* can reach 97.63% while the WVP drops to 11.4 × 10^−14^ g cm/(cm^2^ s Pa) [235].

Three-components composite films, based on PLA mixed with chitosan and loaded with tea polyphenols, in various molar rapports have been obtained by Ye, J. et al. [236]. The three-component film has superior mechanical properties, lower WVP, and better heat-sealability. While the pure PLA has a strength of heat-seal of 2.75 ± 0.41 N/15 mm, the composite film with PLA-tea polyphenols-chitosan presents a strength up to three times higher. The large number of -OH and -NH_2_ moieties from the chitosan polymeric chains and its viscosity ensures an easy and durable welding of the membrane. The tests made on cherries have indicated that the optimum ratio between tea polyphenols and chitosan is 3:7 for prolonging the shelf life from 2 to 8 days at room temperature. In this case was observed the lowest degradation speed and was obtained the best preservation of the vitamins and total solid substance. The vitamin C content was higher for all samples packed with films with tea polyphenols as these substances have antioxidant activity and thus protect vitamin C from interacting with free radicals, decreasing the decay speed.

Like other polymers used for food packaging, the PLA films can also be reinforced with nanoparticles, which can exhibit antimicrobial activity [237]. Zhang, C. et al. have added up to 10% *w/w* AgNPs into PLA obtaining a film that has the capacity to prolong freshness of fruits (strawberries). The optimum results were reported for the film containing 5% AgNPs [150]. Li, W. et al. [238] have reported that both TiO_2_ and Ag nanoparticles can be embedded into PLA. The obtained films have lower WVP when compared with the bare polymer, lower transparency, and lower elastic modulus. The FTIR analysis has failed to evidence any new bonds, indicating the physical embedding of nanoparticles into the PLA. Same results were obtained by DSC, as the vitreous transition and melting temperatures remained unchanged. The only observable modification is the crystallinity degree due to the presence of nanoparticles. Although there is only a physical embedding, the migration tests revealed a good nanocomposite stability, with no detectable levels of nanoparticles migrating into the food. The antimicrobial activity has been tested against *E. coli* and *L. monocytogenes* as typical microorganisms infecting food. As the TiO_2_ percent increased from 1 to 5% the antimicrobial activity improved, the bacterial concentration decreased from 9.12 to 3.45 log CFU/mL. By adding the AgNPs along with TiO_2_ the bacterial concentration further decreased to 3.06 log CFU/mL. Antibacterial activity against *E. coli* for PLA-based films reinforced with aluminum-doped ZnO nanoparticles was reported by Valerini, D. et al. [239] and PLA films loaded with ZnO nanoparticles were investigated for preserving fresh cut apple in [240].

Gelatin films have also been tested in conjunction with various antimicrobial agents as active packaging materials [157,167,241]. In order to slow the release kinetics of the antimicrobials, the active substance can be encapsulated in a copolymer, in beta-cyclodextrin, in clays nanoparticles (like montmorillonite, halloysite etc.) or can be included as a nanoemulsion [241,242]. In order to improve the mechanical and barrier properties, gelatin can be reinforced with other polymers like polycaprolactone [133], sodium alginate [243], chitosan [244], starch [245], cellulose, and derivatives [246]. The mechanical properties are improved also by adding nanoparticles like ZnO [163], TiO_2_ [247], Ag [248].

The bionanocomposites based on poly-(butylene adipate-co-terephthalate)/ Ag_2_O (PBAT/Ag_2_O) obtained directly from solvent (CHCl_3_) have presented promising properties for antimicrobial packaging. The best properties were obtained for 1–10% Ag_2_O load on the PBAT film. Over 10% Ag_2_O the nanoparticles present a tendency to form agglomerates. The mechanical properties of composites were superior to the matrix polymer film up to 7% Ag_2_O. Most importantly, the permeability of oxygen and water vapor was lower for the nanocomposite films than for the simple PBAT film. The antimicrobial activity against *Klebsiella pneumonia* and *S. aureus* strains indicate that these nanocomposite films can be used in food industry [249]. Nisin can also be incorporated into PBAT polymeric matrix to produce an antimicrobial packaging [250], with good results against *L. monocytogenes*.

## 4. Toxicity Studies

The materials used for the innovative antimicrobial food packaging films are usually biopolymers that are biocompatible and safe for human use. Most of them are classified as GRAS and are edible. For example, chitosan or cellulose are considered GRAS in micron scale. While nanocellulose indicate minimal cytotoxicity it still may have an impact over gut microbial population and alter intestinal function, by reducing nutrient absorption [251,252,253]. The use of nanomaterials in food packaging brings a series of advantages as direct antimicrobial activity, support for other antibacterial agents or sensors to detect food contaminants (mostly metabolic products of bacterial activity), but at the same time there is a need to evaluate the risks implied by the use of metallic and oxide nanoparticles. The nanoparticles interact with bacterial cells and can even inhibit biofilms formation, but at the same time can migrate into food and interact with human cells [254,255,256,257]. Therefore, before marketing any novel antimicrobial food packaging, a careful study of nanoparticles migration into food and their associated toxicology must be undertaken. From the available literature studies, it is clear that nanoparticles present toxicological effects in living organisms and may have a profound impact on environmental ecosystems [258,259,260].

AgNPs are among the most studied and used antibacterial agents and therefore many research are devoted to the AgNPs effects and implications for human health [261,262,263]. Studies on zebrafish indicate that AgNPs and AuNPs enhance the secretion of alanine aminotransferase and aspartate aminotransferase. The increased levels of these enzymes lead to increased level generation of ROS, that can cause oxidative stress and immunotoxicity. AgNPs also induced formation of micronuclei and nuclear abnormalities [264]. Unfortunately, with the increased application of nanomaterials, metal nanoparticles have been widely detected in the environment. There are alarming reports about nanoparticles accumulation in seafood [265,266,267], but also about their cytotoxic and genotoxic effect in plants [268,269,270].

Toxicological concerns exist also for copper and copper oxide nanoparticles as studies indicate a strong relation between decreasing nanoparticles size and increasing phytotoxicity [271,272,273]. Some recent studies also indicate effects on plastids, mitochondria, protoplasm, and membranes for *Hordeum sativum distichum* [274] or *Abelmoschus esculentus* [275].

ZnO and TiO_2_ nanoparticles are widely used not only in antimicrobial packaging but also in cosmetics and paints. ZnO is classified by FDA as GRAS and its toxicological effects are still under investigation, but some reports on ZnO nanoparticles are already alarming [276,277,278]. ZnO nanoparticles can pass the stomach and enter in the digestion process in intestine [279] and can induce hyperproliferation of malignant cells [280]. Studies made on mice indicated that TiO_2_ nanoparticles alter gastrointestinal homeostasis, induce cytotoxicity and genotoxicity [281,282,283,284,285].

For essential oils the studies report no toxicity at the concentration used in food packaging [286,287,288]. There are some concerns that at inhibitory concentration for pathogenic bacteria the essential oil can negatively impact also the beneficial bacteria [289].

## 5. Conclusions

As we presented here, there are many strategies that can be adopted to combat the food loss. The use of antimicrobial packaging is gaining ground as both regulatory bodies authorities and general public are taking steps toward acceptance. The general trend is to implement new biodegradable polymers so both problems (antimicrobial activity and plastic pollution) can be addressed. Emphasis is on natural, abundant, and cheap polymers, but they require modification and carefully chosen additives in order to confer the properties required by the packaging industry.

Beside the main polymer, the composite usually contains at least one additive as a plasticizer. This additive plays an important role in achieving the desired mechanical properties. Researchers have a wide range of compounds to choose from: xylitol, mannitol, glycerol, sucrose, polyethylene glycol (PEG), sorbitol, fatty acids, urea, corn syrup, soy proteins, etc. Plasticizers can represent a weight percent between 10 and 65%, depending on the polymer rigidity. Beside the plasticizers, other additives might be included into the polymeric mix, such as antioxidants, antimicrobials, emulsifiers, nutraceuticals, probiotics, etc. Sometimes these additives influence the mechanical properties. While nanoparticles are also reinforcing agents that usually increase the mechanical resistance, essential oils increase the elasticity of the film. Some additives modify the barrier properties, like essential oils or other lipids that decrease WVP.

The replacement of traditional non-compostable oil-derived plastics with antimicrobial biodegradable packaging materials brought new challenges due to incorporation of the antimicrobials into the polymeric matrix, compatibilities between various components and easier degradation by heat and light. Some of the antimicrobials suffer a high loss rate because of inherent volatilization (essential oils for example) and future studies are required to improve the durability and efficiency of novel antimicrobial packaging materials.

The increased demand for cleaner and less processed food, with lower additives content also led to the development of antimicrobial packaging. Beyond the desired antimicrobial activity, the additives used in these innovative packaging can be responsible for the contamination of the food. Migration of undesirable substances must be under the limits established by regulations to ensure the consumer’s safety. For nanoparticles specific toxicological test must be carried out to indicate whether they are safe or not for humans in long term. Also, one must take into account the possible long-term environmental impact of these antimicrobial substances once they end up to landfill. Therefore, reducing the toxicity and environmental impact of synthetic antimicrobial agents is an important subject for further research.

The presence of various additives into the polymeric film must also not affect in a negative way the organoleptic properties of the food, otherwise consumers will reject the new packaging. The packaging must be as transparent as possible because the customers want to see what they are buying. Wrong tints that will give the impression of dusty, dirty, or oxidized will not be received well. Once this kind of antimicrobial packaging is available at a larger scale, the extension of shelf life will become a reality, and since the producers usually mark only the expiry date, customers will not be able to judge the exact freshness of the food. Nevertheless, both regulatory bodies and consumer pressure will shape the future of antimicrobial packaging toward novel, cost efficient, bio-degradable materials that can ensure food safety, quality, and longer shelf life with fewer additives.

## Figures and Tables

**Figure 1 foods-09-01438-f001:**
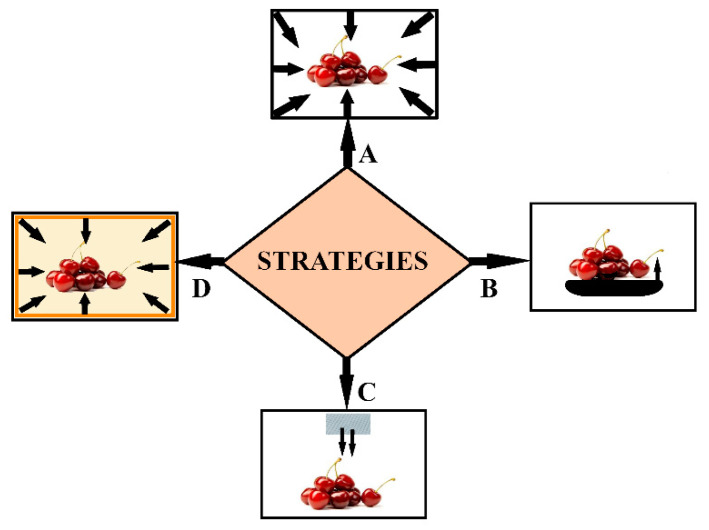
Main strategies employed to obtain antimicrobial packaging. The antimicrobial agents can be: incorporated into the polymeric film (**A**); in direct contact from pads (**B**); released from sachets (**C**); or coated onto polymeric film (**D**).

**Figure 2 foods-09-01438-f002:**
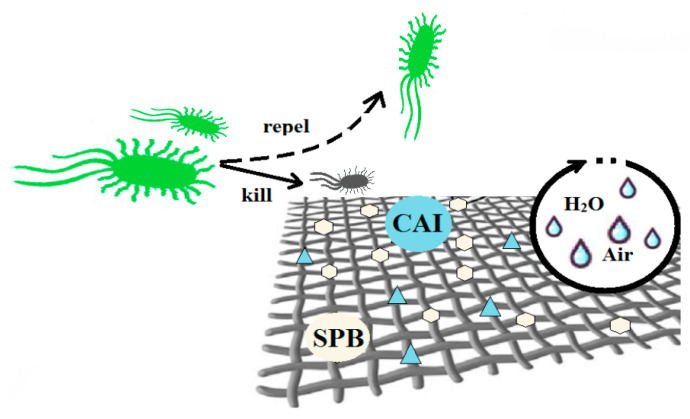
Antimicrobial textile loaded with nanoparticles or other bactericidal and antimycotic agents (isocyanate moiety—CAI and sulfopropyl betaine—SPB).

**Figure 3 foods-09-01438-f003:**
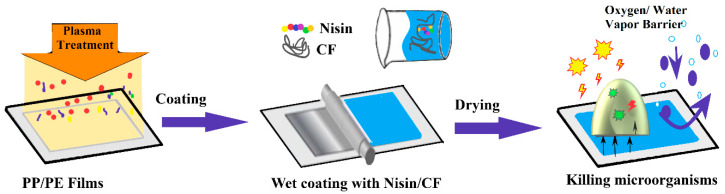
Coating of polypropylene/polyethylene (PP/PE) films with other polymers (cellulose fibers—CF) and loading with antibacterial agents (nisin)—adapted after information presented in [104].

**Figure 4 foods-09-01438-f004:**
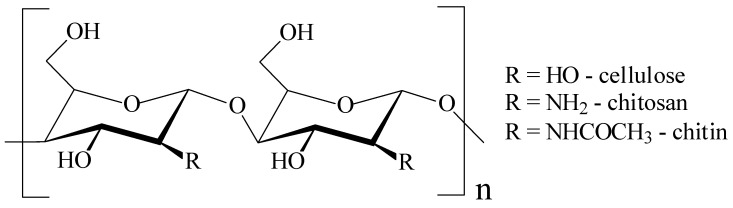
Comparison among structures of various polysaccharides.

**Figure 5 foods-09-01438-f005:**
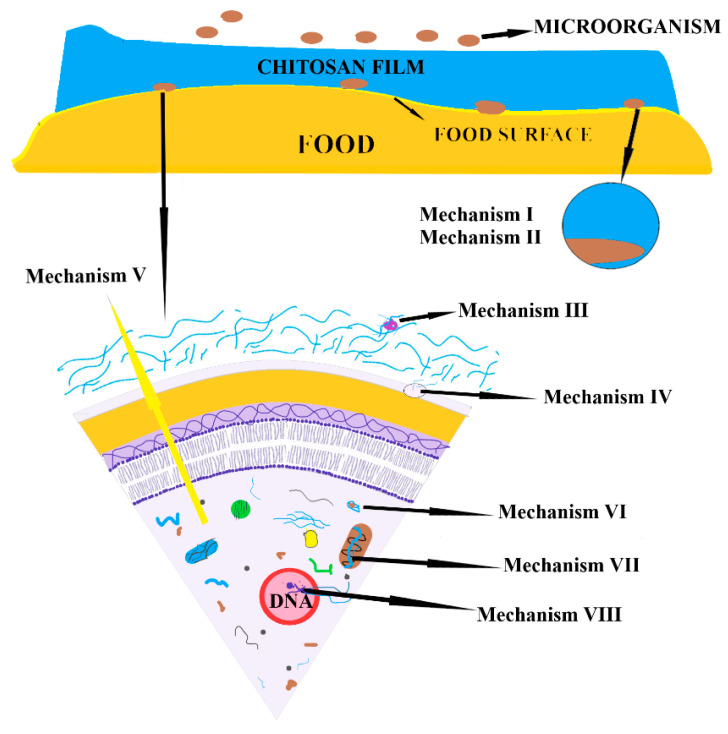
The mechanisms by which the chitosan films manifest antibacterial activity—adapted after information presented in [182].

**Figure 6 foods-09-01438-f006:**
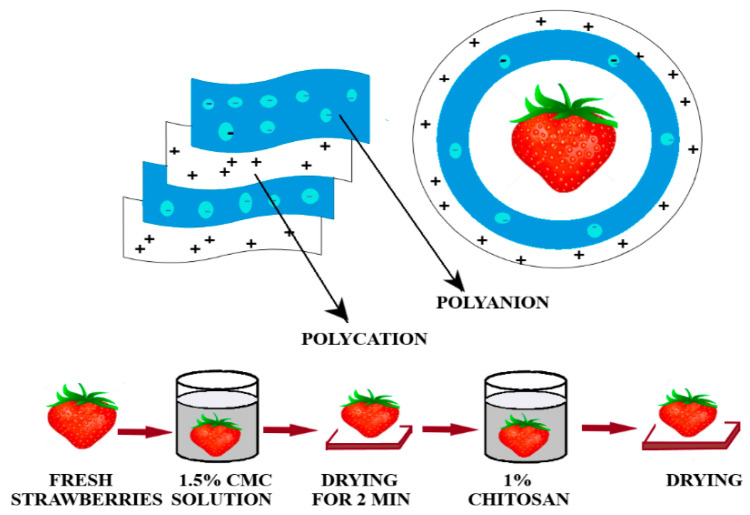
Obtaining the LBL films from chitosan and carboxymethyl cellulose solutions—adapted after information presented in [155].

**Figure 7 foods-09-01438-f007:**
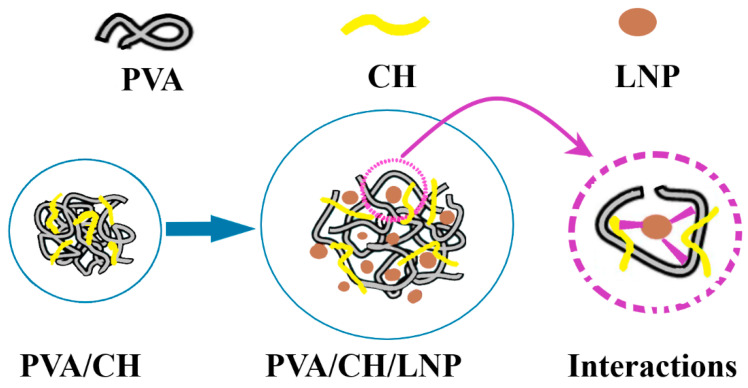
Schematic depiction of polyvinyl alcohol (PVA) and chitosan chains interactions with lignin nanoparticles—adapted after information presented in [194].

**Figure 8 foods-09-01438-f008:**
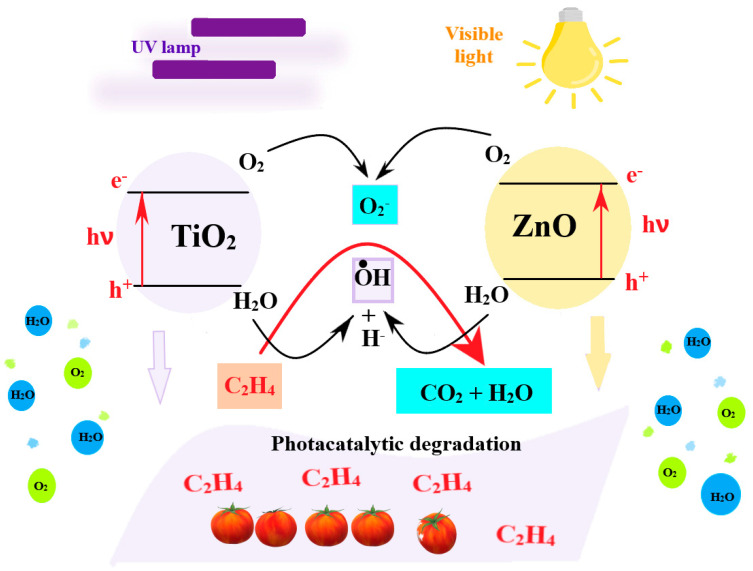
Scheme of the ethylene degradation due to photocatalytic activity of oxidic nanoparticles embedded into chitosan film—adapted after information presented in [154].

**Figure 9 foods-09-01438-f009:**
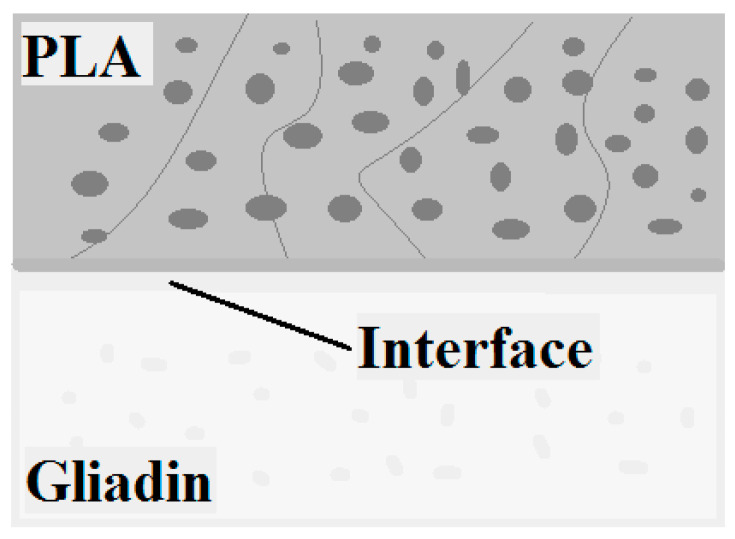
The interface between polylactic acid (PLA) and gliadin layers—adapted after information presented in [158].

**Table 1 foods-09-01438-t001:** Principal film forming biodegradable classes of substances used in packaging.

Film Forming Class	Substance	Reference
Polysaccharides	Cellulose	[110,111]
	Chitosan	[112,113]
	Starch	[114,115]
	Hemicellulose	[116,117]
	Alginate	[118,119]
	Agarose	[120,121]
	Glucomannan	[122,123]
	Pullulan	[124,125]
Proteins	Soy-protein	[126,127]
	Whey-protein	[128,129]
	Zein	[126,130]
	Collagen	[131,132]
	Gelatine	[133,134]
	Casein	[135,136]
Lipids and waxes	Plant oil and animal fat	[137,138]
	Waxes	[139,140]
Others	PLA	[141,142]
	PVA	[143,144]

**Table 2 foods-09-01438-t002:** Antimicrobial packages and their applicability.

Product Preserved	Packaging Material	Antimicrobial Agent	Reference
Iceberg lettuce	Cellulose	Clove and oregano oils	[91]
Cucumber	Chitosan	Limonene	[153]
Tomato	Chitosan	TiO_2_ nanoparticles	[154]
Strawberries	Chitosan/CMC	Chitosan/citric acid	[155]
Strawberries	PLA	AgNPs	[150]
Strawberries	Chitosan/CMC	*Mentha spicata* oil	[156]
Strawberries	Gelatin	Butylated hydroxyanisole	[157]
Fish	PLA	Thymol	[158]
Crap fillets	Alginate/CMC	*Ziziphora clinopodioides* oil / ZnO	[159]
Rainbow trout fillet	Chitosan	Grape seed extract	[160]
Salmon	PLA	Glycerol monolaurate	[161]
Shrimps	Chitosan	Carvacrol	[162]
Shrimps	Gelatin	ZnO /clove oil	[163]
Chicken	Chitosan	Acerola residue extract	[164]
Poultry	Chitosan	Ginger oil	[165,166]
Chicken	Gelatin	Thyme oil	[167]
Chicken	Pullulan	Nisin	[125]
Chicken	Chitosan/PET	Plantaricin	[168]
Ostrich meat	Kefiran/polyurethane	*Zataria multiflora* oil	[169]
Lamb meat	Chitosan	*Satureja* plant oil	[170]
Ground beef	PLA/NC	*Mentha piperita, Bunium percicum*	[171]
Ham	Chitosan/starch	Gallic acid	[172]
Salami	Whey protein	*Cinnamomum cassia*, *Rosmarinus officinalis* oils	[173]
Cheese	Chitosan/PVA	TiO_2_	[174]
Cheese	Cellulose /Chitosan	Monolaurin	[175]
Cheese	Starch	Clove leaf oil	[176]
Cheese	Agar	Enterocin	[177]
Cheese	Zein	Pomegranate peel extract	[178]
Peanuts (roasted)	Banana flour (starch)	Garlic essential oil	[179]

**Table 3 foods-09-01438-t003:** Essential oils and extracts and the usual compatibilities.

Food Type	Essential Oil / Extract	References
Meat	Acerola extract, oregano oil, ginger oil, betel oil, *mentha spicata* oil, rosemarin oil, *ziziphora clinopodioides* oil, *zataria multiflora* oil, *satureja* plant oil, thyme oil, grape seed extract, *eucalyptus* oil	[151,159,160,162,164,165,166,167,169,170,171,173,200,201,202]
Vegetables and fruits	*Litsea cubeba* oil, clove and oregano oils, *mentha spicata*, cinnamon oil, limonene, thyme oil	[91,114,153,156,178,199,203,204]
Pastry and bread	Garlic essential oil, pomegranate peel extract, cinnamon oil, clove oil	[178,179,205]
Dairy and cheese	Clove leaf oil, ginger oil	[107,126,176]

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
