# Peer review of "Biodegradable Antimicrobial Food Packaging: Trends and Perspectives"

_foods, 2020, doi:10.3390/foods9101438_

Round 1

Reviewer 1 Report

The review by Motelica et al describes a perspective on the research trends and possible solutions from recent years in the field of antimicrobial packaging materials. The topic is interesting and the use of innovative antimicrobial packaging is a promising strategy that has attracted great interest from food industries in order to improve the food quality and safety of their products. The authors evaluate the state of the art concerning the implementation of novel biodegradable polymers and the selection of antibacterial and antifungal agents possibly used to develop new antimicrobial polymers for food packaging.

In the paper, many issues are treated, like natural, abundant, and cheap packaging materials, antimicrobial agents from natural sources, polymer activation processes, etc., but in some sections, the subjects have been described in a quite superficial way and on the basis of a limited literature review. In addition, the text in some paragraphs is overly repetitive and exceedingly wordy and there are some grammatical issues and weaknesses in written English.

 Major points

Introduction

The referencing is not always complete. There are statements including only one reference (see as example Ref 4, 5 , 6, 7 etc), and more recently published works should be cited.

 Paragraph “Antimicrobial packaging obtained by modification of current materials”

In this paragraph, the authors describe the main strategies concerning the antimicrobial packaging obtained by modification of current materials. However, a little attention is devoted to the possible drawbacks of each approach and it is not clear what is the real potential of the reported strategies in terms of industrial application. In addition, further research should be taken into account on a new class of promising natural antimicrobial agents such as antimicrobial peptides which can represent a challenge for future applications in food packaging. Finally, no toxicity study reported in the literature has been cited, considering that natural is not necessarily safe.

Author Response

The valuable input from the reviewers to improve the quality of this manuscript is greatly appreciated. The authors have considered and addressed all suggestions.

Point 1: Introduction

The referencing is not always complete. There are statements including only one reference (see as example Ref 4, 5 , 6, 7 etc), and more recently published works should be cited.

Response 1: We have surveyed the literature and recent papers (mostly from 2020) have been indicated as bibliography to sustain statements with insufficient references. Therefore, references: 14-19, 20-25, 28, 29, 31-33 have been added to manuscript.

Point 2: Paragraph “Antimicrobial packaging obtained by modification of current materials”

In this paragraph, the authors describe the main strategies concerning the antimicrobial packaging obtained by modification of current materials. However, a little attention is devoted to the possible drawbacks of each approach and it is not clear what is the real potential of the reported strategies in terms of industrial application.

Response 2: We are thankful for pointing out this weakness in our manuscript. We have added explanations for the attractiveness of this strategy. Briefly the economic costs are lower when minimal modification to production lines is required. The surface modification of existing packaging solution can be done in the last stage of fabrication so there is no need to scrap the entire production line. The main draw back is the continued use of polluting plastics. As regulatory bodies will enforce phasing out the fossil-based plastics, the surface modification, coating solution etc. can be transferred to biopolymeric films. New paragraphs have been added to the section 2 (rows 142-144; 192-205, 252-256)

Point 3: In addition, further research should be taken into account on a new class of promising natural antimicrobial agents such as antimicrobial peptides which can represent a challenge for future applications in food packaging.

Response 3: We thank to the esteem reviewer for the opportunity to enrich the paper by adding relevant results from literature about antimicrobial peptides used in food packaging. Relevant paragraphs with this information were introduced at appropriate places in manuscript together with 10 more references. Rows 192-205, table 2 and rows 636-638

Point 4: Finally, no toxicity study reported in the literature has been cited, considering that natural is not necessarily safe.

Response 4: We are thankful for the suggestion to improve the manuscript. A “toxicity studies” section was introduced to present the concerns raised by the use of some materials. Section 4 was added, rows 730-765

Reviewer 2 Report

The current review is good written, has 209 references cited with recent sources mostly dated by 2018. The review has a high potential to reach researchers and industrial personnael working in the research and development of antimicrobial food packaging.

The abstract summarises all relevant to the review data. In the introduction different approaches of the antimicrobials application for food packaging are schematically represented.

The review is mainly focused on the application of edible and biodegradable packaging materials or last trends in scientific research. At the same time about four pages are devoted to the application of antimicrobials in plain oil sourced packaging materials. There are many technologies and papers devoted to plain packaging with antimicrobial properties which are not highlighted here. Taken this in account I recommend to rename the review to "Antimicrobial food packaging with a target application for compostable packaging: trends and perspectives." Or "Compostable antimicrobial food packaging...", actually PVA mentioned in the review is compostable/biodegradable.

In the conclusion Trends are given and because of the name of the manuscript the reviewer wants to see a short paragraph about future Perspectives. 

In line 9: remove additional 3.

Author Response

The valuable input from the reviewers to improve the quality of this manuscript is greatly appreciated. The authors have considered and addressed all suggestions.

Point 1: The review is mainly focused on the application of edible and biodegradable packaging materials or last trends in scientific research. At the same time about four pages are devoted to the application of antimicrobials in plain oil sourced packaging materials. There are many technologies and papers devoted to plain packaging with antimicrobial properties which are not highlighted here.

Response 1: We have added few other antimicrobial agents and proposed packaging solutions from recent years. Despite our efforts to survey the literature, there might be few relevant articles not covered in present review.

Point 2: Taken this in account I recommend to rename the review to "Antimicrobial food packaging with a target application for compostable packaging: trends and perspectives." Or "Compostable antimicrobial food packaging...", actually PVA mentioned in the review is compostable/biodegradable.

Response 2: The title was changed in “Biodegradable antimicrobial food packaging: trends and perspectives"

Point 3: In the conclusion Trends are given and because of the name of the manuscript the reviewer wants to see a short paragraph about future Perspectives. 

Response 3: We are thankful for the suggestion to improve the manuscript. Some perspectives and future research directions have been added to Conclusions. Rows 783-790, 796-797, 804-806.

Point 4: In line 9: remove additional 3.

Response 4: We have corrected the mistake.

Round 2

Reviewer 1 Report

The authors have addressed all my comments/suggestions. I found their responses quite satisfactory and the revised version has been much improved. I now recommend the paper for publication in Foods journal. Some minor spell checks are required.